# Successional, spatial, and seasonal changes in seed rain in the Atlantic forest of southern Bahia, Brazil

Daniel Piotto[1]*, Dylan Craven[2], Florencia Montagnini[3], Mark Ashton[3], Chadwick Oliver[3], William Wayt Thomas[4]

**1** Centro de Formação em Ciências Agroflorestais, Universidade Federal do Sul da Bahia, Ilhéus, Bahia, Brazil, **2** Biodiversity, Macroecology & Biogeography, University of Goettingen, Göttingen, Germany, **3** School of Forestry and Environmental Studies, Yale University, New Haven, Connecticut, United States of America, **4** The New York Botanical Garden, Bronx, New York, United States of America

* daniel.piotto@ufsb.edu.br

**Data Availability Statement:** All relevant data are within the paper and its Supporting Information files.

**Funding:** DP received funding from Compton Foundation, the Tropical Resources Institute (TRI),

## Abstract

Seed arrival is a limiting factor for the regeneration of diverse tropical forests and may be an important mechanism that drives patterns of tree species' distribution. Here we quantify spatial and seasonal variation in seed rain of secondary forests in southern Bahia, Brazil. We also examine whether secondary forest age enhances seed dispersal and whether seed rain density and diversity in secondary forests decay with distance from mature forest. Across a chronosequence of 15 pairs of mature and secondary forests, 105 seed traps were installed and monitored for one year. We tested the effects of secondary forest age, distance from mature forest, and seasonality on monthly seed rain density, diversity, seed dispersal mode, and diaspore size. We found that secondary forest age had strong, positive effects on the diversity of seed rain, which was generally higher during the wet season. Moreover, contrasting patterns among diversity indices revealed that seeds of rare species occurred more often in 40 yr old secondary forests and mature forests. While the proportion of biotically and abiotically dispersed seeds did not change significantly with distance from mature forest across all forest age classes, we found that biotically dispersed seeds contributed disproportionately more to seed rain diversity. Our results emphasize the importance of biotic dispersal to enhance diversity during secondary succession and suggest that changes in secondary forest structure have the potential to enhance the diversity of tropical secondary forests, principally by increasing dispersal of rare species.

## Introduction

Much of the world's remaining forests are degraded or secondary forests [1]. Secondary forests provide timber and non-timber forest products, protect soils, cycle nutrients, store carbon, maintain watershed functions, and provide reservoirs for biodiversity [2–4]. Yet how well these secondary forests restore ecosystem function and recover biodiversity depends strongly

the Yale Institute of Biospheric Studies "Center for Field Ecology", Garden Club of America, and The Lewis B. Cullman fellowship in tropical environmental biology. WWT received funding from The Beneficia Foundation, the National Science Foundation (DEB 0516233 and 0946618).

**Competing interests:** The authors have declared that no competing interests exist.

on the initial state of land degradation, landscape configuration and connectivity, and local environmental conditions [5, 6].

Seed arrival through seed rain is considered among the most important factors limiting regeneration of tropical secondary forests [7–9]. In the wet tropics, the principal dispersal vectors in open areas are birds and bats that disperse generally small, light seeds of pioneer species, but their proportions vary widely, while dispersal by wind and gravity play a smaller role [10]. Although some studies in rainforests reported a greater input of wind-dispersed seeds from mature forests into young secondary forests than from other sources [11], many studies have shown overwhelming dominance of animal dispersal [10, 12]. For instance, [13] reported a greater input of seeds dispersed by birds and bats than by wind in early successional forests in the Venezuelan Amazon. Nonetheless, a study in Costa Rica found that seed dispersal limitations of small-seeded pioneer species rapidly disappeared following the emergence of young forests (naturally or planted) yet large-seeded, animal-dispersed species remained nearly absent [14].

Distance from seed source plays a vital role in mediating the rate at which secondary forests recover. Forest composition and structure in early stages of secondary forest development in fragmented landscapes are strongly affected by the distance from mature forests, which frequently serve as propagule sources [15]. At the landscape level, previous studies have shown that forest regrowth rates are likely to decrease with distance from remnant forests [16, 17]. Similarly, proximity to mature forests has been found to promote local variation in the structure and composition of secondary forests [18, 19].

Many studies of secondary succession following land abandonment in fragmented landscapes in the Neotropics indicate that the number of dispersed tree seeds is negatively related to the distance to seed sources and perching sites [20–26]). There is also evidence that the distance decay of seed rain from remnant tropical forests depends on seed dispersal syndrome; vertebrate-dispersed seed rain has been found to decrease with distance from remnant tropical forests to a greater extent than wind-dispersed seed rain [11, 12, 27, 28]. However, seed dispersal patterns observed in pastures and agricultural fields may differ from those in secondary forests because of the rapid changes in structure (e.g. canopy height, stratification, and complexity) and in environmental conditions during succession [29]. For example, increases in seed size and abundance of animal-dispersed species during secondary succession [30] likely reflect changes in forest structure that favor shade tolerant species [31, 32].

Seed rain variability is typically correlated with weather at both inter-annual and intra-annual time scales [33]. In tropical forests, precipitation is the most important driver of seed rain because it triggers seed dispersal by enhancing germination conditions [34]. Although moist and wet tropical forests do not show marked fruit production periods like tropical dry forests, a less pronounced but significant intra-annual variation in seed rain abundance and in the relative proportion of biotic and abiotic dispersal has been reported in these forests [35–38].

In this study, we quantify multiple drivers of variation in seed rain during secondary succession in the Atlantic forest of southern Bahia, a center of endemism and high biodiversity of plants and animals [39, 40]. Although comparisons of seed rain between mature and secondary forest or forest and non-forest habitats have been previously reported [27, 41], we are not aware of studies that have examined changes in seed rain along a secondary forest chronosequence in tropical moist forests (but see [42] for tropical dry forests). Here, we used pairs of mature and secondary forests to examine whether changes in secondary forest age enhance seed dispersal and whether seed rain density, diversity, and seed characteristics (e.g. dispersal mode, diaspore size) decay in secondary forests with distance from mature forest. We hypothesize that seed rain density, diversity, and proportions of biotically-dispersed and of large

diaspores will: i) increase with secondary forest age, ii) decrease with distance from mature forest, and iii) vary across seasons.

## Materials and methods

### Site description

The study was conducted in the Serra do Conduru State Park, southern Bahia, Brazil (14°25' south and 39°05' west). The park encompasses about 10,000 ha, has high forest cover ($>$ 80%), and is a mosaic of forest patches in different successional stages, including remnants of undisturbed mature forest. The study sites are at an elevation of 120–300 m asl. The mean annual temperature is 24°C, with annual average precipitation of 2000 mm evenly distributed throughout the year. While the average monthly precipitation year-round is over 100 mm, the months of March to May tend to have higher rainfall and those from August to November less rainfall [43]. The natural vegetation is classified as tropical moist forest [44], with rolling to undulating topography (10–30% slope). Soils are oxisols with low fertility, high iron content and a high percentage of gravel-sized aggregates, which limit water retention and root penetration [45].

### Sampling design

A chronosequence of secondary forests was used to study spatial and successional changes in seed rain. Sites were selected to minimize variation in environmental conditions, past land-use history, and landscape configuration. A total of 95 secondary forest stands larger than 3 ha and adjacent to a mature forest were found in the study area. Fifteen pairs of mature and adjacent secondary forests were randomly selected to represent three age classes of secondary forest: 11 yr (10–12 yr old), 24 yr (22–25 yr old), and 40 yr (37–43 yr old). All sampled secondary forests had been cleared and burned followed by 1–2 years of manioc cultivation. The estimation of site age and past land use was based on a sequence of available aerial photographs and remote sensing data from 1965 to 2009, which provided precise and verifiable estimates of site age. In addition, information on past land use and time of abandonment was verified by interviewing local farmers.

### Seed rain

In January 2009, 105 seed traps were established in mature and secondary forests. In each pair of mature and secondary forests, seven traps were installed at 20 meter intervals. The first trap was placed in mature forest 20 m from the edge, the second trap was set at the edge of the mature forest, and the other five traps were placed at 20 m intervals along a 100 m transect extending into the secondary forest (away from mature forest). From February 2009 to January 2010 material from the seed traps was collected monthly.

Seed traps consisted of a cone of fine-mesh cloth (2mm) attached to a 1m$^2$ circular wire structure suspended on 50 cm legs. Traps were assembled in the field and installed without causing damage to nearby plants. The material collected in the seed traps was dried to a constant weight and carefully separated into diaspores (seeds and fruits), leaves, and twigs. Seeds and fruits collected in each trap were counted, with the exception of species that exceeded 200 seeds per trap. For these species, counts were estimated by weighing a subsample of 200 seeds and extrapolating the total number of seeds using the sample weight. A fruit was considered a single diaspore regardless of its number of seeds for practicality, because separating and counting tiny seeds of some species such as *Cecropia spp*, *Ficus spp*. (biotic dispersal) and *Tibouchina spp*. (abiotic dispersal) proved to be impractical. Damaged seeds and fruits were included in

the counting. Diaspores were classified to morphospecies and, when possible, to the lowest taxonomic level (genus or species) by comparing them to herbarium specimens and the seed collection of the Centro de Pesquisas do Cacao (CEPEC) herbarium in Ilhéus, Bahia and relevant literature [46, 47]. Diaspores were measured, photographed, and classified into five size classes following [36, 48]: ≤3 mm, >3 and ≤6 mm, >6 and ≤15 mm, >15 and ≤30 mm, and >30 mm length. Based on diaspore morphology, morphospecies were classified into two primary dispersal modes: biotically dispersed species, which displayed diaspores with fleshy pulp, aril, or other features associated with animal dispersal agents; and abiotically dispersed species (wind, barochory or explosive dispersal), which displayed winged propagules and plumes or are dispersed by free fall or by explosive dehiscent fruits. Seed predation and removal from traps was likely minimal given the high sampling frequency [11, 27] and it was considered that the effect of loss or damage is equal across the temporal and spatial gradient.

## Seed rain diversity

We quantified diversity of seed rain using diversity orders 0 and 2 expressed in effective species numbers to facilitate interpretation [49, 50], which correspond to species richness and Simpson diversity, respectively. Species richness emphasizes rare species, while Simpson diversity gives greater weight to highly abundant species. From here on, diversity orders 0 and 2 are referred to as $^0D$ and $^2D$. Because of small sample sizes at the seed trap level, species diversity estimates, particularly species richness, were not considered to be reliable [51] and, therefore, were not analyzed. For this reason, we pooled data by forest age class, month, and distance from mature forest in order to make robust coverage-based rarefaction and extrapolation estimates of diversity for species richness and Simpson diversity [52]. Additionally, we calculated coverage-based estimates of species richness and Simpson diversity for each seed dispersal mode (abiotic or biotic) across secondary forest age classes, distance from mature forest, and months. All estimates of species diversity were made using the 'iNext' R package [53] in the R statistical computing language, version 3.5 [54].

## Data analysis

We used generalized linear mixed effects models to test the effects of month, distance, and secondary forest age on monthly seed rain density, seed dispersal mode, and diaspore size. Data from mature forests were excluded from these analyses, so as not to conflate the effects of secondary forest age with distance from mature forest. In all models, time was coded as a categorical variable where each month was a level, as there was no *a priori* expectation that temporal changes in any response variable would be linear. Secondary forest age class (11, 24, and 40 yr old) also was treated as a categorical variable, while distance from mature forest was treated as a continuous variable. Study site was initially included in all models as a random intercept.

For seed rain density, we fit a generalized linear mixed effects model with a Poisson distribution and site- and observation-level random effects to account for overdispersion [55]. We included mean seed rain density and bootstrapped 95% confidence intervals of the mature forests in figures for the purposes of comparison but they were not included in any model. To examine changes in seed dispersal modes, i.e. seeds dispersed biotically or abiotically, we fit a generalized linear mixed effects model with a binomial distribution, a logit link, and site- and observation-level random effects. Here, the response variable was modeled as the log-odds ratio of seeds to the total amount of seeds, which was weighted by the total number of seeds. Lastly, we assessed diaspore size using a generalized linear mixed effects model with a binomial distribution and a logit link. Given the ecological importance of large diaspores in mature tropical forests yet their relative low frequency in secondary forests, we modeled diaspore size

as the log-odds ratio of large diaspores (here classified as being larger than 15 mm) to the total amount of diaspores. Because our initial model for diaspore size had a singular fit, i.e model parameters were on the boundary of parameter space and could not be used to estimate model coefficients robustly, we simplified the model by removing the site-level random effect. For both seed dispersal mode and diaspore size, we present model results in terms of estimated probabilities, i.e. the probability that seeds will be dispersed abiotically or the probability that diaspores will be larger than 15 mm. We estimated statistical significance of model parameters using Wald tests with the 'car' package. All mixed effects models were fitted using 'lme4' [56]. For purposes of qualitative comparison, we included mean seed rain density, proportion of abiotically dispersed seeds, and proportion of large seeds in the mature forests and their boot-strapped 95% confidence intervals. For coverage-based estimates of species diversity, differences across forest age classes, distance from mature forest, and months were assessed by examining 95% confidence intervals for overlap [52]. Analyses were performed in the R statistical computing language, version 3.5 [54]. All data necessary to replicate these analyses is provided in the Supplementary Information (S1 Table).

### Authorization for the field work

The study was carried out in the Serra do Conduru State Park. The authorization to carry out the field work inside the park was provided by INEMA (Bahia State Environmental Agency) and the authorization to collect botanical material was provided by the Instituto Chico Mendes de Conservação da Biodiversidade-ICMBio/MMA (SISBIO n. 20591–1).

## Results

A total of 26,581 diaspores of 274 morphospecies were captured in seed traps from February 2009 to January 2010. Seed rain density (diaspores $m^{-2}$ $month^{-1}$) increased significantly with age in secondary forests and was highest in the mature forests (Fig 1a; Table 1). However, seed dispersal mode did not vary significantly with forest age (Table 1; Fig 2a). There was a steady increase in seed rain species richness with forest age (S1a Fig). Seed rain species richness in the 40 yr old secondary forests and the mature forests was very similar (95% confidence intervals overlapped) and was greater than in the 11 and 24 yr old secondary forests (S1a Fig). In contrast, Simpson diversity of seed rain was not affected by forest age (S1a Fig). During secondary succession, species richness and Simpson diversity of biotically dispersed species was consistently higher than that of abiotically dispersed species across forest age classes, with distance from mature forest, and over time (Fig 3). Across secondary forest age classes, we observed a steady increase of species richness of biotically dispersed seeds but that of abiotically dispersed seeds did not change markedly (Fig 3a). This pattern was not consistent across measures of species diversity; Simpson diversity of biotically dispersed seeds reached its maximal values in 24 and 40 yr old forests, while values for abiotically dispersed seeds were highest in 11 and 24 yr old forests (Fig 3b).

While the proportions of abiotically and biotically dispersed seeds were similar across forest age classes (Table 1; Fig 2a), diaspore size differed significantly (Fig 4a). In the 11 yr old and the 40 yr old forests, there were significantly higher probabilities of larger diaspores occurring than in the 24 yr old forests (Table 1; Fig 4a). However, mature forests appear to have a larger proportion of larger diaspores than any of the secondary forests (Fig 4a). In general, small diaspores, e.g. diaspores in ">3 and ≤6 mm" and ">6 and ≤15 mm" size classes, were usually the most abundant size class across all forest age classes (S2 Fig).

Seed rain density decreased significantly with increasing distance from mature forests (Table 1; Fig 1b). Species richness and Simpson diversity of seed rain, however, varied non-

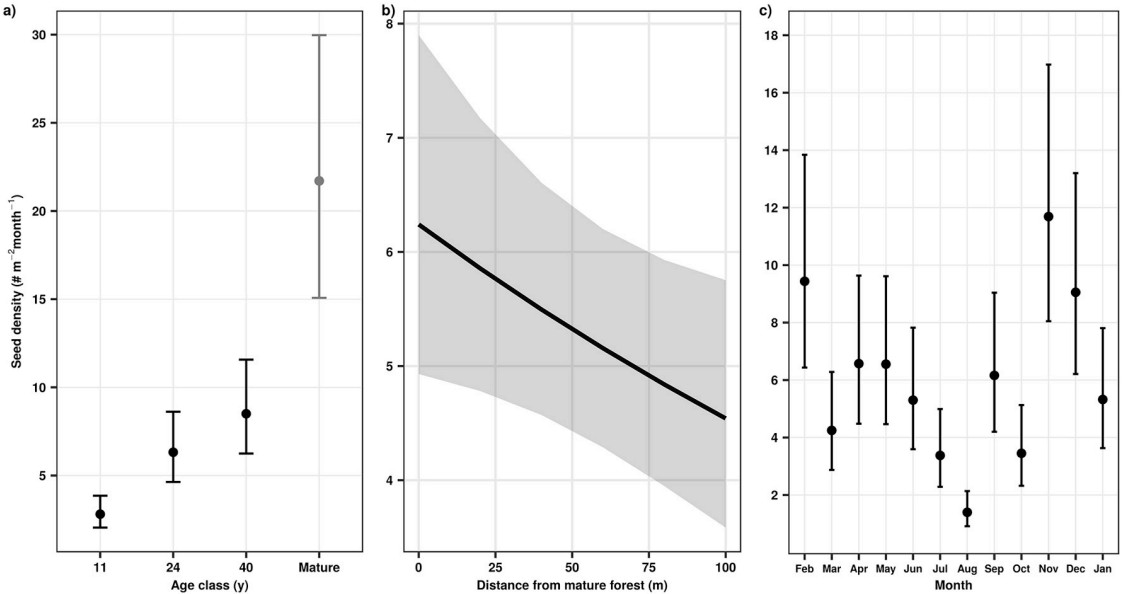

**Fig 1. Successional, spatial, and seasonal changes in seed rain density in tropical secondary forests of southern Bahia, Brazil.** Estimates and confidence intervals are from a generalized linear mixed effects model with a Poisson distribution. Whisker bars are 95% confidence intervals; in b, gray band indicates 95% confidence intervals and black line indicates that the fixed effect is statistically significant (p ≤ .05). In a, we included mean seed rain in mature forests for comparison, but these observations were not estimated by the model (see Methods); the presented value is mean seed rain density in mature forests and whisker bars are bootstrapped 95% confidence intervals. See Table 1 for model fit information.

linearly with distance from mature forest (S1b Fig). Similarly, species richness of biotically and abiotically dispersed seeds did not change consistently with distance to mature forest (Fig 3c). In contrast, Simpson diversity of biotically dispersed species was initially high at the edge of mature forests and then rapidly declined with distance from mature forests (Fig 3d). Simpson diversity of abiotically dispersed seeds increased non-linearly from the edge of mature forests, where it was lowest, to its maximal values, 100 m from the edge of mature forests (Fig 3d). Seed dispersal mode did not vary significantly with distance from mature forests (Table 1; Fig 2b), nor did diaspore size (Table 1; Fig 4b).

Seed rain density, as well as species richness and Simpson diversity of seed rain showed marked seasonal changes (Table 1; S1c Fig). The lowest estimate of seed rain density was observed in August and the highest in November (Fig 1c). Species richness and Simpson diversity of biotically dispersed species exhibited greater temporal variation than that of abiotically

**Table 1. Summary of generalized linear mixed-effects models evaluating successional, spatial, and seasonal changes in seed rain, seed dispersal mode, and diaspore size in tropical secondary forests of southern Bahia, Brazil.**

| Fixed Effects | Seed density (#/m2/month) | Seed dispersal mode | Diaspore size |
|---|---|---|---|
| Forest age (y) | *25.96* | 1.93 | *16.05* |
| Distance from mature forest (m) | *4.27* | 0.40 | 0.73 |
| Month | *98.66* | *146.87* | *102.60* |
| Marginal R$^2$ (%) | 16.37 | 14.24 | 2.14 |
| Conditional R$^2$ (%) | 98.81 | 73.01 | 11.36 |

Marginal and conditional R$^2$ represent model variation, explained by fixed effects and the combination of fixed and random effects, respectively. For fixed effects, values are χ$^2$ statistics; bold and italicized values are statistically significant at α = 0.05.

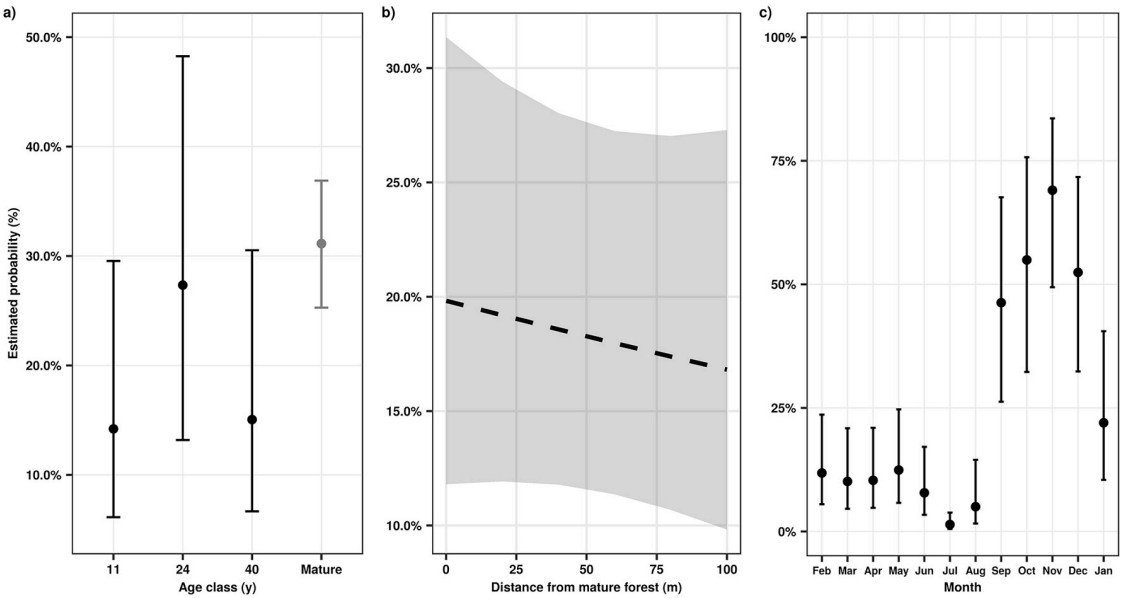

**Fig 2. Successional, spatial, and seasonal changes in seed dispersal mode in tropical secondary forests of southern Bahia, Brazil.**
Estimated probability is the probability that seeds are dispersed abiotically. Estimates and 95% confidence intervals are from a generalized linear mixed effects model with a binomial distribution. Whisker bars are 95% confidence intervals; in b, gray band indicates 95% confidence intervals and dashed line indicates that the fixed effect is not statistically significant (p ≤ .05). In a, we included mean proportion of abiotically dispersed seeds in mature forests for comparison, but these observations were not estimated by the model (see Methods); the presented value is the mean proportion of abiotically dispersed seeds in mature forests and whisker bars are bootstrapped 95% confidence intervals. See Table 1 for model fit information.

dispersed species (Fig 3e and 3f). Seed dispersal mode also varied significantly with time (Table 1); abiotically dispersed species were more common between September and December than during the rest of the year (Fig 3c). Diaspore size also exhibited significant seasonal variation (Table 1); large diaspores were very infrequent from June through January, after which they became more abundant (Fig 4c), but they were always less numerous than smaller diaspores.

## Discussion

Overall, our results indicate that the density and species richness of seed rain increases with forest age in secondary forests. The increase in seed rain density with forest age found in this study is most likely due to autochthonous seed rain, i.e. belonging to species already present in the canopy. With increasing forest age, more species are able to reach the canopy and thus are more likely to produce seeds. Even though it was not possible to determine whether seeds were locally produced or actively dispersed from other locations, most of the seeds collected in the seed traps belonged to locally abundant pioneer species such as *Tibouchina elegans*, *Henriettea succosa*, *Salzmannia arborea*, and *Miconia lurida* [57]. In the same region of the Atlantic forest, [58] found high abundances of the same pioneer species in the seed rain of forest gaps and recently burned sites, and attributed their abundance to autochthonous seed rain. [59, 60] also found that most seeds in secondary forests of the Amazon and Costa Rica belonged to tree species already established in secondary forests. However, successional changes in forest structure did not affect the proportion of seed dispersal modes. In contrast to previous studies, e.g. [30, 61]), we found similar proportions of abiotically and biotically dispersed seeds across forest ages. Our results suggest that, even in younger secondary forests, forest structure is sufficiently

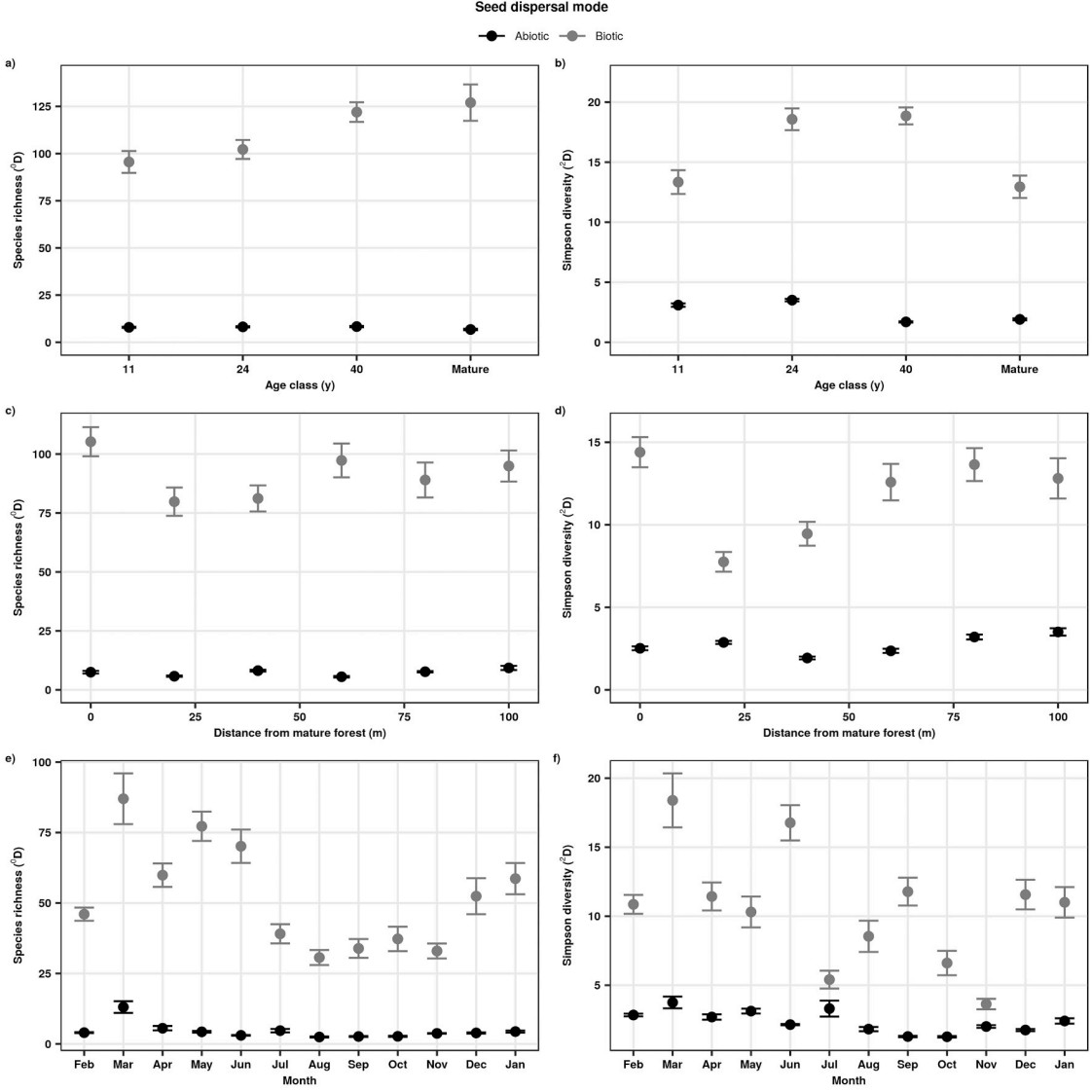

**Fig 3. Successional, spatial, and seasonal changes in the diversity of seeds dispersed abiotically and biotically in tropical secondary forests of southern Bahia, Brazil.** Species diversity was estimated as species richness (a, c, e; Hill number 0 or $^{0}D$) and Simpson diversity (b, d, f; Hill number 2 or $^{2}D$) using sample-coverage based rarefaction and extrapolation. Sample coverage was 97.9% for a and b, 98% for c and d, and 96% for e and f. Whisker bars are 95% confidence intervals.

complex to provide habitat that attracts birds and bats, which are the principal seed dispersers of pioneer- and non-pioneer species in the study region [57, 62].

### Do seed rain density, diversity, and proportions of biotically-dispersed and of large diaspores increase with secondary forest age?

While species richness of seed rain increased with forest age, Simpson diversity, which emphasizes common species, was stable across the secondary forest chronosequence. Our results show that 40 yr old secondary forests and mature forests have more rare species in their seed rain than that of younger secondary forests, a pattern that is consistent with previous studies in Amazonian and Atlantic forests [58, 59]. The greater species richness of seed rain in mature

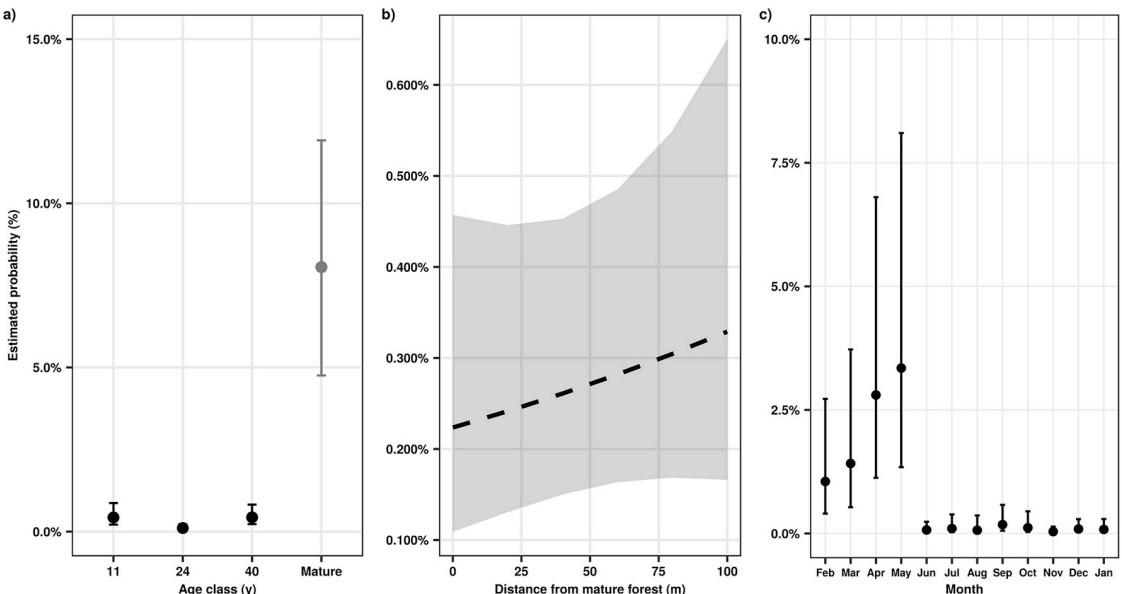

**Fig 4. Successional, spatial, and seasonal changes in diaspore size in tropical secondary forests of southern Bahia, Brazil.**
Estimated probability is the probability that diaspores are larger than 15 mm. Estimates and 95% confidence intervals are from a generalized linear mixed effects model with a binomial distribution. Whisker bars are 95% confidence intervals; in b, gray band indicates 95% confidence intervals and dashed line indicates that the fixed effect is not statistically significant (p ≤ .05). In a, we included mean proportion of large diaspores in mature forests for comparison, but these observations were not estimated by the model (see Methods); the presented value is the mean proportion of diaspores greater than 30 mm in mature forests and whisker bars are bootstrapped 95% confidence intervals. In b, dashed line indicates that the fixed effect is not statistically significant (p ≤ .05). See Table 1 for model fit information).

forests and 40 yr old secondary forests observed in this study is likely related to the divergent contributions of seed dispersal modes. For both diversity indices, seed rain diversity of biotically dispersed seeds was highest in mature forests and 40 yr old secondary forests and was consistently higher than abiotically dispersed seeds in all forest age classes. Thus, our results suggest that seed rain diversity increases during secondary succession because older secondary forests likely attract more animal dispersers, which disperse a large quantity and diversity of seeds.

Forest age also enhanced the dispersal of large seeds, which were less common in secondary forests than in adjacent mature forests. Our results coincide with those reported by [36], who found that forest interior habitat had a significantly higher percentage of medium, large, and very large seeds and more species compared to forest edges in the Brazilian Atlantic forest. Our study and others have shown that young secondary forests are overwhelmingly composed of small-seeded pioneer tree species (many of which are bird and bat dispersed), whereas many large-seeded species that persist in mature forests may be rare or absent in young secondary forests [14, 29, 62–64]. While seed rain diversity may increase with forest age, our results suggest that the seed rain composition in younger tropical secondary forests is unlikely to resemble that of mature forests in the region [65].

### Do seed rain density, diversity and the proportions of biotically-dispersed and of large diaspores decrease with distance from mature forest?

We found that the effect of distance to mature forest on the seed rain of secondary forests was only significant for seed rain density. Similar to previous studies on abandoned pastures and

agricultural fields [20–24, 66–68], the results of this study show that seed rain density is negatively affected by the distance to mature forest, with a higher number of seeds found in secondary forests closer to the edge of mature forests. However, we did not find differences in the proportion of large seeds found along the distance gradient.

Although there is evidence in the literature that the decrease in seed rain density with distance from remnant mature forest is unequal between wind- and vertebrate-dispersed seeds [11, 20, 27, 28, 69], in this study seed dispersal modes did not vary significantly along the distance gradient. This likely reflects that the most prolific seed dispersers in the study region, i.e. birds and bats, are highly vagile and, therefore, are better able to access secondary forests that are relatively far from mature forests [64]. Moreover, our seed traps did not account for direct or secondary seed dispersal of large seeds by terrestrial mammals, which may be more sensitive to the distance to mature forests or habitat complexity of secondary forests than more vagile dispersal vectors.

Tree species with large seeds are usually the most affected by distance to seed sources early in secondary succession, because larger seeds are less abundant and more likely to be dispersed over shorter distances [64, 68]. However, this trend is expected to change during secondary succession because the rapid recovery of tree cover enhances animal dispersal by creating a habitat matrix that facilitates animal movement [61, 70].

We did not find a significant relationship between distance from mature forest and seed rain diversity, which appears to be driven largely by the spatial distribution of rare species (S1b Fig). Species diversity of rare species ($^0D$) varied non-linearly with distance, suggesting that their spatial distribution is stochastic [71]. However, our results show differences in the proportion of large-sized diaspores between secondary and mature forests and a significant decrease in seed rain with distance from mature forest. These findings indicate that colonization dynamics of these secondary forests are influenced by an unequal input of seeds along the distance gradient and by the absence of large-sized seeds in younger secondary forests. The temporal and spatial variation in seed rain observed in this study may be an increasingly important mechanism that drives patterns of tree species' distribution in older secondary forests.

### Do seed rain density, diversity, and proportions of biotically-dispersed and of large diaspores vary across seasons?

Seed rain exhibited significant seasonal variations in both density and diversity. Seed rain species richness peaked during rainy months (March–May) and was very low during dry months (August–November). Likewise, diaspore size also exhibited significant seasonal variation. Large diaspores were very infrequent from June through January, when seed rain richness was very low. Other studies in the Atlantic forest also showed that trees usually disperse fruits during the rainy season [35, 37, 38]. Seed rain seasonality in southern Bahia (central Atlantic forest) is similar to the seed rain seasonality reported by [36] in Pernambuco state (northern Atlantic forest) and by [35] in Parana state (southern Atlantic forest). Both studies reported peaks of seed rain between February and May and low seed rain in July and August. These results suggest that phenological seasonality in seed rain is fairly synchronized along the Atlantic forest and that seed rain may be strongly mediated by intra-annual variation in precipitation. However, these patterns may be affected by inter-annual variation in weather conditions that changes the frequency and size of masting events [72].

Biotically and abiotically dispersed species exhibited contrasting phenological seasonality. While abiotically dispersed species have clear peaks in seed rain density in dry months (Sept—Dec), the diversity of abiotically dispersed seeds is consistently low throughout the year. The

contrast in seed rain density between biotic and abiotic dispersal is likely due to differences in seed dormancy. Previous studies have found that seeds dispersed during the dry season are mostly orthodox seeds, which show a delay between seed dispersal and germination [35, 73]. In this study, biotically dispersed seeds appear to drive patterns of seed rain diversity to a greater extent than abiotically dispersed ones, as they contribute more rare and common species. Diversity patterns did not respond to the significantly higher density of abiotically dispersed seeds during dry months. This shows that there is a limit to the capacity of abiotically dispersed seeds to enhance seed rain diversity in regenerating forests, thus emphasizing that gains in diversity of seed rain are predominantly facilitated by the establishment of forest structure that facilitates biotic seed dispersal. However, the impact of increases in seed rain density and diversity due to successional changes in forest structure on the composition and diversity of tree seedling communities may be attenuated by numerous non-random barriers to recruitment in successional forests [7, 41].

## Conclusions

In the present study, we show that secondary forest age, distance from mature forest, and seasonality affect seed rain. Our results show that successional changes, which increase habitat complexity for animal seed dispersers, strongly enhance seed rain density and diversity in tropical secondary forests in a biodiversity hotspot. Importantly, seeds of rare species were more abundant in the oldest secondary forests and mature forests. Our analysis also highlights the disproportionate contributions of biotically dispersed seeds to seed rain diversity, which become particularly apparent during the wet season. While seed rain density predictably declines with distance from mature forest, seed rain diversity did not, suggesting that the dominant biotic seed dispersers in this study region–birds and bats–move readily within secondary forests. Together, our results emphasize the importance of biotic dispersal to enhance diversity during secondary succession in tropical forests. However, the low amount of large seeds across forest age classes likely indicates that factors other than habitat provided by secondary and mature forests, such as landscape connectivity and hunting pressure, limit dispersal by large seed dispersers. As secondary forests become increasingly important for ecosystem provisioning and biodiversity conservation in tropical regions, it is important to gain a better understanding of the factors that enhance or constrain dispersal within and across human-impacted landscapes.

## Supporting information

**S1 Fig. Successional, spatial, and seasonal changes in diversity of monthly seed rain in tropical secondary forests of southern Bahia, Brazil.** Diversity estimates were made using sample coverage-based rarefaction and extrapolation. Sample coverage was 99.3% for a), 96.5% for b), and 98.8% for c). Whisker bars are 95% confidence intervals.
(TIFF)

**S2 Fig. Seed size class distribution by dispersal modes across forest age classes in tropical secondary forests of southern Bahia, Brazil.** Proportion of seed class sizes was calculated by pooling one year of seed rain density across forest age classes.
(TIFF)

**S1 Table. Seed rain data of secondary forests of southern Bahia, Brazil.**
(TXT)

## Acknowledgments

We thank the staff of Instituto Floresta Viva, Serra do Conduru State Park, and Comissão Executiva do Plano da Lavoura Cacaueira (Ceplac). We also thank L. Romero, V. da Silva, and C. Viana for their assistance with the field and lab work.

## Author Contributions

**Conceptualization:** Daniel Piotto, Florencia Montagnini, Mark Ashton, Chadwick Oliver, William Wayt Thomas.

**Formal analysis:** Daniel Piotto, Dylan Craven.

**Funding acquisition:** Daniel Piotto, William Wayt Thomas.

**Investigation:** Daniel Piotto.

**Methodology:** Daniel Piotto, Dylan Craven.

**Supervision:** Florencia Montagnini, Mark Ashton, Chadwick Oliver, William Wayt Thomas.

**Writing – original draft:** Daniel Piotto, Dylan Craven.

**Writing – review & editing:** Florencia Montagnini, Mark Ashton, Chadwick Oliver, William Wayt Thomas.

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
