## [Decision Letter · Decision Letter 0]

16 Oct 2019

PONE-D-19-21585

Successional, spatial, and seasonal changes in seed rain in the Atlantic forest of southern Bahia, Brazil

PLOS ONE

Dear Prof. Piotto,

Thank you for submitting your manuscript to PLOS ONE. After careful consideration, we feel that it has merit but does not fully meet PLOS ONE’s publication criteria as it currently stands. Therefore, we invite you to submit a revised version of the manuscript that addresses the points raised during the review process.

A reviewer has recommended accepting this manuscript for publication, pending minor changes. I agree with his(her) assessment and am willing to consider a revised version for publication in the journal, assuming that you are able to modify the manuscript according to the recommendations.

We would appreciate receiving your revised manuscript by Nov 30 2019 11:59PM. To enhance the reproducibility of your results, we recommend that if applicable you deposit your laboratory protocols in protocols.io, where a protocol can be assigned its own identifier (DOI) such that it can be cited independently in the future. For instructions see: http://journals.plos.org/plosone/s/submission-guidelines#loc-laboratory-protocols

We look forward to receiving your revised manuscript.

Kind regards,

Angelina Martínez-Yrízar

Academic Editor

PLOS ONE

Journal Requirements:

2. We note that Figure S1  in your submission contains a copyrighted image. All PLOS content is published under the Creative Commons Attribution License (CC BY 4.0), which means that the manuscript, images, and Supporting Information files will be freely available online, and any third party is permitted to access, download, copy, distribute, and use these materials in any way, even commercially, with proper attribution. For more information, see our copyright guidelines: http://journals.plos.org/plosone/s/licenses-and-copyright.

a)     You may seek permission from the original copyright holder of Figure S1 to publish the content specifically under the CC BY 4.0 license.

Additional Editor Comments:

A reviewer has recommended accepting this manuscript for publication, pending minor changes. I agree with his(her) assessment and am willing to consider a revised version for publication in the journal, assuming that you are able to modify the manuscript according to the recommendations.

Reviewers' comments:

Reviewer's Responses to Questions

**Comments to the Author**

1. Is the manuscript technically sound, and do the data support the conclusions?

Reviewer #1: Yes

2. Has the statistical analysis been performed appropriately and rigorously? 

Reviewer #1: Yes

3. Have the authors made all data underlying the findings in their manuscript fully available?

Reviewer #1: No

4. Is the manuscript presented in an intelligible fashion and written in standard English?

Reviewer #1: Yes

5. Review Comments to the Author

Reviewer #1: This is an impressive study examining the effects of secondary forest age, distance from mature forest, and season on seed rain (density and diversity). The study appears to be well conducted, they collected an impressive amount of data (26,500 diaspores from 15 plots over 12 months) and, for the most part, it is well written. Most of my comments are about the methods (some parts need to be explained in more detail) and discussion.

1. My main critique is that the Discussion is difficult to follow and some of the results are not really discussed (see comments 11 and 12 below). This is a challenging study to discuss in that the authors collected data on several factors, so they have A LOT of results to discuss, and several of them don’t follow easily-interpretable patterns.

2. L107: “during the first 40 years of succession” This sounds like the authors collected data over 40 years. Perhaps it can be reworded to clarify that secondary forests up to 40 years old were examined. (Same issue in line 30 in the abstract.)

3. L122: “haplorthox” is not regularly-used jargon. Perhaps provide more info about what this term means.

4. L130: How did you determine the age of the secondary forests?

5. L144: “For these species, counts were estimated with subsamples.” Please explain in more detail.

6. L145: “A fruit was considered a single diaspore regardless of its number of seeds.” Please explain your rationale. This method seems like it would underestimate fleshy-fruit (and animal) dispersal compared to dry seed (and wind) dispersal.

7. L153: “animal dispersed species, which displayed features associated with vertebrate dispersal agents” Please state your criteria for labeling a diaspore as animal-dispersed versus wind-dispersed.

8. L192-193: I think it makes more sense to re-word this as: “we modeled diaspore size as the log-odds ratio of large diaspores (here classified as being larger than 15 mm) to the total amount of diaspores”

9. L184-185 and L200-202: When you say “we included [means and 95% confidence intervals]” does that mean you included them in the results/graphs? When I first read these sentences, I thought you meant that these means were included in your models, and it was very confusing. Please clarify.

10. L251: It would be more accurate to say “our results indicate that the density and species richness of seed rain increases with forest age”

11. L320: “this study did not show a significant relationship between distance from mature forest and seed rain diversity” This is a key result that is not discussed. It’s contrary to your hypothesis stated in the Intro. Any ideas why? Are your results similar to or different from previous studies?

12. The pattern in Figure 3d seems surprising. Why is the diversity of biotically-dispersed diaspores highest at the edge of the mature forest, but then it drops at 20 m, before steadily climbing with distance from the mature forest? I would have expected it to DECREASE with distance from mature forest. Any possible explanations?

6. PLOS authors have the option to publish the peer review history of their article (what does this mean?). If published, this will include your full peer review and any attached files.

Reviewer #1: No

---

## [Author Response · Author response to Decision Letter 0]

8 Nov 2019

Below we present the verbatim comments of editor and reviewer followed by our response in capital letters with a description of changes we have made to the manuscript.

RESPONSE TO THE EDITOR

DONE.

2. We note that Figure S1 in your submission contains a copyrighted image. We require you to either (1) present written permission from the copyright holder to publish these figures specifically under the CC BY 4.0 license, or (2) remove the figures from your submission:

BECAUSE OF COPYRIGHT ISSUES, WE DECIDED TO EXCLUDE FIGURE S1 FROM SUPPORTING INFORMATION.

3. We note that you have indicated that data from this study are available upon request. PLOS only allows data to be available upon request if there are legal or ethical restrictions on sharing data publicly.

BECAUSE THERE ARE NO RESTRICTIONS, WE HAVE UPLOADED THE DATASET USED IN THIS STUDY AS SUPPORTING INFORMATION.

RESPONSE TO REVIEWERS

1. My main critique is that the Discussion is difficult to follow and some of the results are not really discussed (see comments 11 and 12 below). This is a challenging study to discuss in that the authors collected data on several factors, so they have A LOT of results to discuss, and several of them don’t follow easily-interpretable patterns.

WE THANK THE REVIEWER FOR ACKNOWLEDGING THE COMPLEXITY OF THIS STUDY AND FOR THE THOROUGHLY REVIEW. WE ADDRESSED COMMENTS 11 AND 12 TO IMPROVE THE DISCUSSION SESSION OF THE MANUSCRIPT.

2. L107: “during the first 40 years of succession” This sounds like the authors collected data over 40 years. Perhaps it can be reworded to clarify that secondary forests up to 40 years old were examined. (Same issue in line 30 in the abstract.)

DONE. WE REPRHASED L107 AND L30.

3. L122: “haplorthox” is not regularly-used jargon. Perhaps provide more info about what this term means.

DONE. WE DELETED THE SOIL SUB-ORDER (HAPLORTHOX) BUT WE DECIDED TO LEAVE THE SOIL ORDER (OXISOLS) WHICH IS COMMONLY USED IN THE SOIL LITERATURE.

4. L130: How did you determine the age of the secondary forests?

DONE. WE PROVIDED DETAILED INFORMATION ABOUT THE WAY AGE OF SECONDARY FORESTS WAS DETERMINED.

5. L144: “For these species, counts were estimated with subsamples.” Please explain in more detail.

DONE. WE PROVIDED DETAILED INFORMATION ABOUT THE SEED COUNTING METHOD.

6. L145: “A fruit was considered a single diaspore regardless of its number of seeds.” Please explain your rationale. This method seems like it would underestimate fleshy-fruit (and animal) dispersal compared to dry seed (and wind) dispersal.

A FRUIT WAS CONSIDERED A SINGLE DIASPORE FOR PRACTICALITY. WE CLARIFIED THAT IN THE METHODS BY PROVIDING SOME EXAMPLES OF BIOTIC AND ABIOTIC DISPERSED SPECIES THAT HAVE EXTREMELY TINY SEEDS, FOR WHICH THE PROCESS OF SEPARATION AND COUNTING PROVED TO BE IMPRACTICAL.

7. L153: “animal dispersed species, which displayed features associated with vertebrate dispersal agents” Please state your criteria for labeling a diaspore as animal-dispersed versus wind-dispersed.

DONE. WE PROVIDED MORE DETAILS ABOUT THE FEATURES USED TO DETERMINE DISPERSAL MODE.

8. L192-193: I think it makes more sense to re-word this as: “we modeled diaspore size as the log-odds ratio of large diaspores (here classified as being larger than 15 mm) to the total amount of diaspores”

DONE. WE REWORDED THIS SENTENCE.

9. L184-185 and L200-202: When you say “we included [means and 95% confidence intervals]” does that mean you included them in the results/graphs? When I first read these sentences, I thought you meant that these means were included in your models, and it was very confusing. Please clarify.

DONE. WE CLARIFIED THAT WE JUST INCLUDED MEAN SEED RAIN AND BOOTSPTRAPPED 95% CONFIDENCE INTERVALS OF MATURE FORESTS IN THE FIGURES BUT THEY WERE NOT INCLUDED IN ANY MODEL.

10. L251: It would be more accurate to say “our results indicate that the density and species richness of seed rain increases with forest age”

DONE. WE CHANGED ‘DIVERSITY’ FOR ‘SPECIES RICHNESS’ AS SUGGESTED.

11. L320: “this study did not show a significant relationship between distance from mature forest and seed rain diversity” This is a key result that is not discussed. It’s contrary to your hypothesis stated in the Intro. Any ideas why? Are your results similar to or different from previous studies?

THANKS FOR POINTING THIS OUT. THIS RESULT REFLECTS THAT SPATIAL PATTERNS OF RARE SPECIES ARE STOCHASTIC (I.E. THEY DO NOT VARY DETERMINISTICALLY WITH DISTANCE) AND THAT COMMON SPECIES ARE SIMILARLY ABUNDANT ACROSS THE SPATIAL GRADIENT. WE CHANGED THE FIRST SENTENCES OF THIS PARAGRAPH (STARTING AT L320) TO CLARIFY THIS PART OF THE DISCUSSION. WE INCLUDED A NEW REFERENCE ABOUT THE STOCHASTICITY OF SEED DISPERSAL, BUT WE DID NOT COMPARE OUR RESULTS WITH OTHER SIMILAR STUDIES BECAUSE WE ARE UNAWARE OF STUDIES THAT HAVE USED A SIMILAR EXPERIMENTAL DESIGN (PAIRS OF MATURE AND SECONDARY FORESTS AND SEED TRAPS PLACED AT 20M INTERVALS ALONG A 100M TRANSECT).

12. The pattern in Figure 3d seems surprising. Why is the diversity of biotically-dispersed diaspores highest at the edge of the mature forest, but then it drops at 20 m, before steadily climbing with distance from the mature forest? I would have expected it to DECREASE with distance from mature forest. Any possible explanations?

YES, WE ALSO EXPECTED THAT SPECIES DIVERSITY (0D and 2D) WOULD DECREASE WITH DISTANCE FROM MATURE FOREST. THE LOW SPECIES DIVERSITY OF BIOTICALLY-DISPERSED SEEDS (2D) AT 20 M AND 40 M FROM THE MATURE FORESTS SUGGESTS THAT MANY OF THE SEEDS BELONGED TO RARE AND NOT COMMON SPECIES (COMPARE TO FIG. 3C). WE HAVE NO MECHANISTIC EXPLANATION FOR THIS PATTERN, AND THINK THAT THIS UNEXPECTED RESULT SIMPLY REFLECTS THE STOCHASTICITY OF SPATIAL PATTERNS OF SEED DISPERSAL.

---

## [Editor Report · Decision Letter 1]

2 Dec 2019

Successional, spatial, and seasonal changes in seed rain in the Atlantic forest of southern Bahia, Brazil

PONE-D-19-21585R1

Dear Dr. Piotto,

We are pleased to inform you that your manuscript has been judged scientifically suitable for publication and will be formally accepted for publication once it complies with all outstanding technical requirements.

With kind regards,

Angelina Martínez-Yrízar

Academic Editor

PLOS ONE
---

## [Editor Report · Acceptance letter]

9 Dec 2019

PONE-D-19-21585R1 

Successional, spatial, and seasonal changes in seed rain in the Atlantic forest of southern Bahia, Brazil 

Dear Dr. Piotto:

I am pleased to inform you that your manuscript has been deemed suitable for publication in PLOS ONE. Congratulations! Your manuscript is now with our production department. 

With kind regards,

on behalf of

Dr. Angelina Martínez-Yrízar 

Academic Editor

PLOS ONE